# Gesture-Informed Robot Assistance
# via Foundation Models

**Li-Heng Lin**[1], **Yuchen Cui**[1†], **Yilun Hao**[1], **Fei Xia**[2], **Dorsa Sadigh**[1*]

**Abstract:** Gestures are a fundamental and significant mode of non-verbal communication among humans. Deictic gestures (e.g. pointing), in particular, offer valuable means of efficiently expressing intent in situations where language is inaccessible, restricted, or highly specialized. As a result, it is essential for robots to comprehend gestures in order to infer human intentions and establish more effective coordination with them. Prior work often rely on a rigid hand-coded library of gestures along with their meanings. However, interpretation of gestures is often context-dependent, requiring more flexibility and common-sense reasoning. In this work, we propose a framework, GIRAF, for more flexibly interpreting gesture and language instructions by leveraging the power of large language models. Our framework is able to accurately infer human intent and contextualize the meaning of their gestures for more effective human-robot collaboration. We instantiate the framework for interpreting deictic gestures in table-top manipulation tasks and demonstrate that it is both effective and preferred by users, achieving 70% higher success rates than the baseline. We further demonstrate GIRAF's ability on reasoning about diverse types of gestures by curating a *GestureInstruct* dataset consisting of 36 different task scenarios. GIRAF achieved 81% success rate on finding the correct plan for tasks in *GestureInstruct*. Project website: `tinyurl.com/giraf23`

**Keywords:** Planning with Gestures, Human-Robot-Interaction, LLM Reasoning

## 1 Introduction

Gestures are an important form of communication that we frequently use in our everyday activities such as at traffic intersections, restaurants, and department stores. They are used to disambiguate and communicate our intent especially when language is not available or limited (e.g., waving at a car to pass through), or when language is too specialized (e.g. pointing at *hex screwdriver* vs *cross slot screwdriver* instead of referring to them by their name as shown in Fig. 1). The ability to understand human gestures is thus critical for autonomous robots to predict human intent, and be able to respond effectively. While prior work has studied how gestures can be beneficial in human-robot interaction, these approaches are often rigid, requiring extensive engineering effort for predefining a library of fixed gestures accompanied with their corresponding meaning [1, 2, 3, 4]. This not only is expensive, but also assumes a fixed mapping between gestures and their meaning. However, interpretation of gestures can be highly dependent on the context and may require situated reasoning leading to different human intents. For example, pointing at a cup calls for potentially different actions, such as *picking up the cup* or *pouring into the cup* given different contexts and history.

Recent works have demonstrated that foundation models such as large language models (LLMs) trained on internet data have enough context for commonsense reasoning [5, 6, 7, 8], making moral judgements [9, 10, 11], or reward design [12, 13]. Similarly, we expect LLMs to be able to reason about and interpret gestures *under the assumption* that they are prompted with a textual description of the gesture and the context. Addressing this assumption is often referred to as the *grounding* problem. However, grounding gestures is not as simple as recognizing the gesture class (e.g., thumbs-up,

---

* [1] Computer Science Department, Stanford University, Stanford, CA, USA. [2] Google Deepmind.
† Corresponding author. Email: `yuchenc@stanford.edu`

7th Conference on Robot Learning (CoRL 2023), Atlanta, USA.

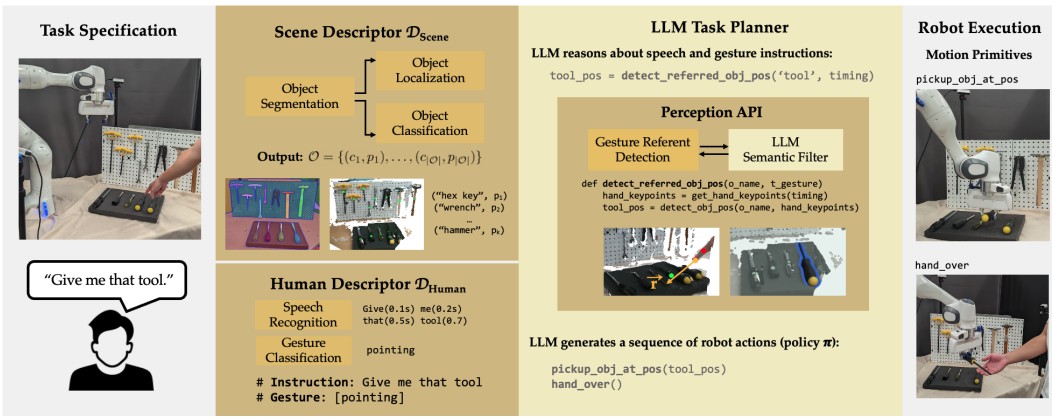

Figure 1: **GIRAF System Diagram.** GIRAF consists of 1) grounding modules including a scene descriptor $D_{\text{Scene}}$, a human descriptor $D_{\text{Human}}$ and a set of perception API; and an LLM task planner that is prompted with few-shot demonstrations to reason about the high-level multimodal task specification and generate code calling functions from other modules. Grounding modules are colored brown, and reasoning modules are colored yellow. As an example, a human user provides language instruction "give me that tool" along with a pointing gesture to specify the particular tool they want. GIRAF process information about the objects in the scene with $D_{\text{Scene}}$ and grounds the human input with $D_{\text{Human}}$. The LLM task planner is prompted to interpret the multimodal instruction and produce policy code that commands the robot to complete the human's request.

waving, or pointing), as it also requires detecting the context the gesture is presented in such as the referent of deictic gestures (i.e. what the person pointing at).

While recent work shows that vision-language models (VLMs) can provide some amount of grounding by describing the scene, these models are limited in two key ways: First, many of them focus on describing the objects but not the state of the human, and thus their context is limited to the scene but not the history of human actions or intents. Second, even if one could extend VLMs to go beyond the scene and describe humans, existing VLMs lack enough geometric reasoning capabilities to accurately interpret gestures. The absence of geometric reasoning in pre-trained VLMs is not surprising since it falls outside the scope of their original objectives. Furthermore, fine-tuning VLMs to capture geometric reasoning about gestures requires considerable amounts of 3D data [14].

Instead of only relying on VLMs to fully reason about gestures, our insight is to leverage a combination of existing pre-trained vision models along with other contextual information such as language instruction to ground LLMs to reason about human gestures and accurately infer their intent. We propose a system, GIRAF (Gesture-Informed Robot Assistance via Foundation-Models), that leverages expert models for identifying gestures, and prompts an LLM for reasoning about gesture-language instructions and generating robot policies. Specifically the LLM will directly generate robot code, for both perceiving and acting, given the language and gesture descriptions. An overview of the system is shown in Fig. 1. In this example, the user wants a tool that they do not know how to describe in language and use gesture to specify what tool the robot should grasp. The user provides a language and gesture specification, in this case "Give me that tool" along with pointing gestures towards the tool to disambiguate the language instruction. Our framework consisting of a *Human Descriptor* and a *Scene Descriptor* detect the pointing gesture and objects in the scene. Then our *LLM Task Planner* reasons about the task and uses a *Perception API* to detect the object being referred by the gesture. The LLM then directly generates motion primitives such as picking up and handing over the item.

To evaluate the effectiveness of our system, we instantiate GIRAF for instruction following in table-top manipulation settings. We first conduct a user study focusing on interpreting *deictic* pointing gestures, showcasing how language alone can be inefficient or ineffective for specifying a task. Our user study shows that users significantly prefer GIRAF over a language-only baseline and are also better at achieving task goals when using GIRAF with more than 70% higher success rates. We then construct a *GestureInstruct* dataset to demonstrate the capability of LLMs to reason about gestures in various contexts in zero- or few-shot manner. GIRAF achieved 81% success rate on finding the correct plan for tasks in *GestureInstruct*.

## 2 Related Work

**Gestures for Human-Robot Interaction.** As an important mode of non-verbal communication, gestures have been studied both within and outside the context of human-robot interaction (HRI). The full literature is out of the scope of this paper and we refer the readers to the survey work of Vuletic et al. [15] on gestures in HRI. In this section we focus on the most relevant literature of interpreting *speech-related* gestures or *co-verbal* gestures [15]. Prior work on incorporating gestures for HRI often relies on defining a fixed set of gesture vocabulary and hand-coding their mappings to robot actions [1, 2, 3, 4]. One major limitation of these rule-based methods is that the human user needs to learn which gestures to use for eliciting which robot response within predefined context. Prior work has also explicitly focused on understanding *deictic gestures* such as pointing. Matuszek et al. [16] and Chen et al. [17] take a data-driven approach for interpreting natural deictic gestures. Whitney et al. [18] propose a multimodal Bayes filter for interpreting referential expressions. These methods are specialized referent detectors that require either large amounts of in-domain training data or extensive engineering effort for each novel task domain. In contrast, GIRAF offloads fusion of multimodal instructions to LLMs and implements a heuristic-based referent detection module leveraging VLMs and simple geometry so that it can perform referent detection in a zero-shot manner.

**LLM-based Planning for Robotics.** Large language models have been adapted as the user interface for various robotics applications. LLMs have shown promising performance in a variety of complex reasoning tasks [5, 6, 10, 12]. Multiple efforts have demonstrated planning capabilities of LLMs in robotics, including zero-shot generation of high-level task plans [19], reasoning about the state of the task from various sources of feedback [20], planning with contextually appropriate and feasible actions [21, 22], and directly producing robot code [23, 24]. These works ground task scenes using VLMs but do not explicitly model the presence of the human beyond providing language instructions and feedback. However, LLMs and VLMs are not pretrained for complex geometric reasoning and therefore these systems fail in tasks where there are multiple semantically similar objects or the user just does not have the right language to describe an item. GIRAF bypasses complex geometric reasoning from language alone by incorporating a human descriptor that grounds human gestures for providing richer feedback for LLM reasoning.

## 3 Gesture-Informed Robot Assistance via Foundation Models

**Problem Statement.** We define the problem of *embodied multimodal instruction following* as: given a task specification in the form of synchronized speech $\mathcal{S}$ and visual gestures $\mathcal{V}$, a robot needs to output a control policy $\pi$ that satisfies this task specification. We assume access to a set of parameterized robot action primitives $\mathcal{A} = \{a^1(\theta_1), \ldots, a^n(\theta_n)\}$. The robot policy $\pi$ needs to return a sequence of action primitives with parameters specifying object locations.

The key insight of our system is that we can leverage LLMs pretrained on large-scale human-generated text corpus for reasoning about such multimodal instructions, and effectively turning the instruction following problem into two sub-problems: 1) the *grounding* problem of translating the raw human input $\mathcal{S}$, $\mathcal{V}$ and scene observations $\mathcal{O}$ into a form that LLMs can consume, and 2) the *prompting* problem of enabling LLMs to reason about the grounded context for producing a robot policy $\pi$.

We propose an LLM-based framework for multimodal instruction following, named gesture-informed robot assistance via foundation models (GIRAF). The system diagram of GIRAF is shown in Fig. 1. In the rest of this section, we discuss how we approach each of the sub problems in multimodal instruction following, and provide implementation details for GIRAF.

### 3.1 Grounding Speech, Gestures, and Scene

For the grounding problem, we introduce different descriptors to separately ground the scene and the human in the scene. The scene descriptor $D_{\text{Scene}}$ provides information about objects while the human descriptor $D_{\text{Human}}$ textualizes the speech instruction and the gesture provided. At the same time, instead of passively detecting all possible objects and gesture-related information and feeding them

to the LLM planner, we prompt the LLM to first reason about the speech and gesture instructions, and then decide which perception tools to use for detecting necessary information. The perception tools include expert gestures models that extract gesture information for different gesture classes.

**Scene Descriptor.** $D_{\text{Scene}}$ is used to describe object categories and locations in the scene. Given an image of the scene, $D_{\text{Scene}}$ performs *Object Segmentation* using Segment Anything[25], and then compute object centers (*Object Localization*) by de-projecting 2D pixels to 3D positions using depth information. We use OpenCLIP [26] to find object labels by comparing the similarity between the embedding of the segmented object image with a list of all possible object labels. This list of all possible labels can be provided by object detectors or the human user. In our experiments, we use ground truth object lists to minimize error introduced by object detectors. (*Object Classification*). The scene descriptor outputs object information as a set $\mathcal{O} = \{(c_1, p_1), \ldots, (c_{|\mathcal{O}|}, p_{|\mathcal{O}|})\}$ where $c_i$ and $p_i$ are the label and 3D position of object $i$ respectively.

**Human Descriptor.** The goal of $D_{\text{Human}}$ is to textualize human speech and gesture. For *speech recognition*, GIRAF processes audio input and turns speech into text with each word marked with its timing. The timing information is used for determining which visual frames to process for aligning with gesture information. We employ Microsoft Azure [27] for this purpose. *Gestures* can be described with different levels of details or fidelity. For example, a "pointing" gesture can also be described as "index finger extends out and others curl inward", or even numeric values of the relative position of each finger tips with respect to the wrist. In addition, gestures can be static (e.g. thumbs-up, pointing at an object) or dynamic (e.g. waving, drawing a circle over a group of objects). It is non-trivial to design a universal gesture detector based on hand information only due to the context-dependant nature of gestures. We find LLMs are capable of reasoning about gesture representations with different level of fidelity but perform best with human-annotated gesture labels when they are available and can be reliably detected (more details in Section 4).

In our instantiation of GIRAF, we employ off-the-shelf MediaPipe [28] hand detector to extract hand features and train deep neural networks to predict a set of selected gesture classes. We train supervised classification models (one for static gestures, another for dyanmic ones) to classify gestures using data from EgoGesture [29]. Specifically, we extract the 1) keypoints of detected hands in image coordinates, 2) keypoints of detected hands in world coordinates, and 3) confidence of detected hands from the output of MediaPipe hand detector and concatenate them as the input feature to gesture classification models. We train separate models with different architectures for classifying static and dynamic gestures. For the static gesture model, we use a three-layer multilayer perceptron (MLP) with ReLu as the activation function. For the dynamic gesture model, we input the preprocessed landmark features into a recurrent neural network (RNN) with an LSTM unit. We then feed the output of the RNN to a linear layer. We find stat-of-the-art VLMs can also provide promising gesture classification results on static gestures but often fail to detect the correct referent (see Appendix F for more details) and therefore we design specialized referent detection module.

The output of $D_{\text{Human}}$ are two lines of code in comment (as shown in the bottom left box of Fig. 1)—one for language instruction, the other for gesture—which we then feed to the LLM task planner.

**Perception API.** In this work, we mainly focus on deictic gestures, where in addition to detecting the gesture, our system needs to detect the *referent* the gesture is referring to. The task planner will decide whether the deictic gesture is referring to an object, a location, or a direction. To find the referred *object*, the referent detection function takes an object name $o_{\text{target}}$ and a timing $t_{\text{gesture}}$ for gesture frame as inputs. We first query *semantic filter* with $o_{\text{target}}$ to get a set of candidate object classes $\mathcal{C}$. We implement *semantic filter* by prompting the LLM to return a list of candidate object names given a label to harness LLM's capability of reasoning about object semantics. For example, when the user says "Give me that tool", $\mathcal{C}$ will only contain labels that can be classified as "tool" (subject to the LLM). We then filter a list of candidate object instances $\bar{\mathcal{O}}$ using $\mathcal{C}$:

$$\bar{\mathcal{O}} = \{(c_k, p_k) | (c_k, p_k) \in \mathcal{O} \text{ and } c_i \in \mathcal{C}\} \tag{1}$$

Next, we index the visual frames using $t_{\text{gesture}}$, leverage MediaPipe [28] to detect hand keypoints, and de-project 2D keypoints to 3D keypoints. We then find the *pointing ray* that starts at the detected

index finger tip $p_{\text{tip}}$ and goes in the direction of the vector from index finger pip joint $p_{\text{pip}}$ to the tip (denoted as $\overrightarrow{r}$ in Fig. 1). Let $\overrightarrow{v}_{\text{finger}}$ denote the unit vector from $p_{\text{pip}}$ to $p_{\text{tip}}$. The pointing ray is $\overrightarrow{r} = (p_{\text{tip}}, \overrightarrow{v}_{\text{finger}})$. To identify the referred object, we use a heuristic that returns the object in $\bar{\mathcal{O}}$ that is closest to the pointing ray:

$$(c_{\text{target}}, p_{\text{target}}) = \underset{(c_i, p_i) \in \bar{\mathcal{O}}}{\arg\min} [D(p_i, \overrightarrow{r})] \tag{2}$$

To find a referred *position*, we follow a similar procedure as above, but instead of considering the positions corresponding to objects, we find the point in the entire point cloud that is closest to $\overrightarrow{r}$. If the user use pointing for referring to a *direction*, we return $\overrightarrow{v}_{\text{finger}}$ as the referred direction.

### 3.2 Prompting LLMs for Task Planning

To adapt LLMs for planning, we take the approach from the work of Liang et al. [23] and directly prompt the LLM to generate language model programs (LMPs) in Python by providing perception APIs, motion primitive APIs, and demonstrating how to use each of these functions. The LLM we use in our experiments is GPT-3.5 (text-davinci-003) with 0 temperature. An example plan generated by LLM for the instruction "give me that tool" with a pointing gesture is shown in Fig. 1. The LLM planner first reasons about the instructions and realizes this is a deictic gesture, therefore leverages perception APIs to locate human hand and the referred object. The LLM then sequences the motion primitives to pick up the detected referent and hand it over to the human. We show below another example plan generated by the LLM planner for a similar task. Here the human breaks up the instruction into two steps and request the robot to pick up the object first and then hand it over. To show that the LLM planner can also reason with gestures of different fidelity level, we use detailed description of the gesture shape instead of a high-level gesture label in this example:

```
# Instruction 0: pick up the water jug
# Gesture: index finger extends out while others curl inward
water_jug_pos = detect_referred_obj_pos('water jug')
pick_up_obj_at_pos(water_jug_pos)

# Instruction 1: hand it to me
# Gesture: an open palm faces upward
target_pos = detect_hand_center_pos()
move_gripper_to_pos(target_pos); open_gripper()
```

Note that we removed code for confirmation and error detection in these examples to highlight LLM's reasoning capability for interpreting gestures. In our implementation of GIRAF, we leverage Microsoft Azure [27] text-to-speech functionality and include examples in the prompt such that the robot would 1) confirm with the user by repeating the instruction it receives, 2) report error if it cannot produce bug-free code or failed to detect a gesture (when it thinks the speech should accompany a gesture), and 3) ask the user if it took a correct first movement to decide whether to continue executing or abort. For example, before picking up anything, the robot will move to the object, "point" at the predicted target object with its gripper, and ask the human if it has located the correct target. If the human responds "no", then the robot will abort the picking up action. This level of transparency adds some overhead to the interaction but asserts that the robot will not make unexpected moves, which is important for ensuring safety when we do not have full control over what the LLM outputs.

## 4 Experiments

### 4.1 Can GIRAF enable instruction following robots to be more natural and efficient?

We hypothesize that incorporating gesture information, especially deictic pointing gestures, would make task specification easier and more natural for the human, outperforming prior language-only approaches to instruction following. Concretely, we make the following hypotheses:

- **H1.** GIRAF is preferred by human users compared to a language-only instruction following method for specifying goals that are ambiguous or hard to describe.
- **H2.** GIRAF is quantitatively more effective compared to the language-only instruction following method for specifying goals that are ambiguous or hard to describe.

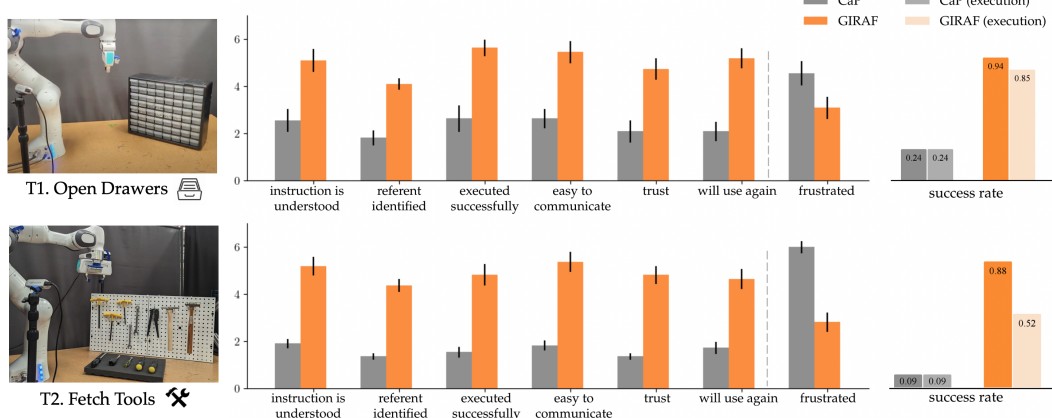

Figure 2: **User Study Results.** GIRAF is rated higher by users and enables better performance than baseline language-only method CaP [23]. Note that for performance we report both planning success rates (identifying the correct referent) and execution success rates.

To test **H1** and **H2**, we conduct a user study with two table-top manipulation tasks where the user instructs a 7-DoF Franka Panda robot arm to manipulate specific objects on the table while there are multiple instances of semantically the same or similar objects in the scene. We compare our method with the baseline language-only method CaP [23]. Specifically, we designed two tasks shown on the left side of Fig. 2:

- **T1: Open Drawers** In this task, there are 64 drawers in a cabinet and the user needs to specify one for the robot to open. Here, we expect language to not be an efficient modality, as a language-only method would require the user to specify the exact row and column of the drawer.
- **T2: Fetch Tools** In this task, a user is asking the the robot the hand over a tool they would like to use. These tools require the human to use specialized language to refer to them, and are additionally difficult for the VLM to classify correctly.

We recruited 11 participants for the user study (6 female, 5 male). 6 out of the 11 participants are native English speakers. Each user participated in the two different task settings as described above. For each task, they select three different target objects, resulting in about 30 tests in total, and for each object, they have at most 3 trials (speech recognition failures do not count towards trials). This study is IRB approved. More details are included in Appendix C and implementation of the motion library is included in Appendix D.

**Gestures provide an easier mode of communication when interacting with robots.**    As shown in Fig. 2, users rated GIRAF higher (except for frustration, which lower is better) than the baseline across our qualitative measures and GIRAF scored highest for "easy to communicate" (supporting **H1** with $p < .01$). We also see that the baseline is rated lower for *fetch tools* task than in *open drawers* task as we expected. Most users could count and describe the position of drawers they want but lack the language to specify the tools.

**Gestures enable higher success rate in instruction following.**    The rightmost plots of Fig. 2 show the success rates of baseline method CaP and GIRAF for both tasks in our user study (supporting **H2** with $p < .01$). We plot both the planning success rate and the execution success rate defined in Appendix C. GIRAF achieves higher success rates on both tasks, and we also find out one major limitation for the baseline to identify the correct object purely from language lying in the fact that existing LLMs are not good at geometric reasoning such as understanding relative positions.

## 4.2   Can GIRAF reason about diverse types of gestures?

We evaluate the LLM planner's capability of reasoning about diverse types of unseen gestures in HRI settings in a zero- or few-shot manner. We curated the dataset *GestureInstruct* with 36 different

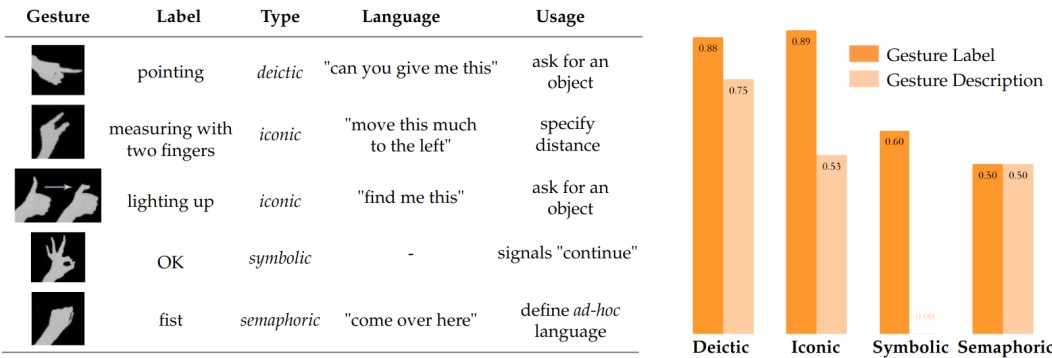

Figure 3: **Reasoning about diverse gestures.** Representative gestures of each type (left). Success Rate of Gesture Reasoning with GIRAF on *GestureInstruct* using different gesture representations (right).

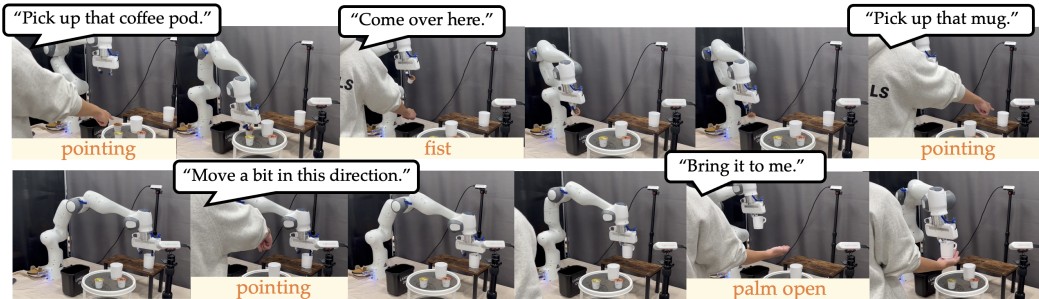

Figure 4: **Long horizon interaction with multiple gestures.** We visualize a long-horizon interaction using GIRAF that involves multiple different gestures unseen in the prompt, and pointing gestures that refer to object, location, or direction depending on context.

speech-gesture task scenarios in human-robot interaction settings (the complete dataset along with annotations is on our website). We generated each task scenario under four different communicative gesture types—*symbolic*, *semaphoric*, *iconic* and *deictic*—and include both static and dyanmic gestures (see details in Appendix B). *Symbolic* gestures are conventional gestures used to convey a fixed meaning and can replace language (e.g. thumbs-up). *Semaphoric* gestures are gestures that human design for specific purposes and associate meaning by telling the robot what it means, then they would use them again like symbolic gestures (e.g. ask the robot to turn around on a snapping gesture). *Deictic* gestures are gestures used to physically refer to an object, a location, or a direction (e.g. pointing with index finger). *Iconic* gestures are gestures that are used to represent objects, actions, intents, and abstract concepts (e.g. use two hands opening and closing to represent book).

Each task scenario in *GestureInstruct* consists of an image or sequences of images showing the visual gesture, a high-level gesture label, an accompanying language instruction (except for symbolic gestures), an interaction context (e.g. robot state and the scene), and a description of the human intent in the form of goal state. An example task scenario in *GestureInstruct* is using *iconic* gesture 'hammering', and ask the robot to "give me the tool that does this" in the context of "robot sees a screwdriver, a wirecutter and a hammer". The LLM task planner needs to infer the correct type of gesture and call the corresponding function to handle detection and grounding of iconic gestures in order to find the correct tool. Additional examples of gesture-speech instruction-following task scenarios under each gesture type are shown in Fig. 3 (left) and the full dataset is on our website.

**GIRAF is able to reason about a diverse set of gestures in zero-shot manner.** Our results show that GIRAF achieves 80.6% success rates across all task scenarios in *GestureInstruct* dataset. Fig. 3 (right: dark orange bars) shows the results under each gesture type and we find that GIRAF performs better on object-related gestures (deictic and iconic) comparing to symbolic or semaphoric gestures. Our conjecture is that both symbolic and semaphoric gestures are not accompanied by language, so the reasoning needs to purely rely on the gesture itself. Since LLM does not actually see the gesture,

its ability to reason about these gestures is limited. We also implement the full pipeline of GIRAF to show its reasoning capabilities for various gestures in a long horizon task. This long-horizon interaction is visualized in Fig. 4, demonstrating that GIRAF is able to reason about symbolic gestures, iconic gestures, and deictic gestures referring to different entities in a single interaction. We provide additional example use cases in the Appendix.

### 4.3 Can GIRAF reason with different representations of gestures?

To understand what is the best way to ground gestures, we investigate what kind of representation enables high reasoning performances. Gestures can be represented differently with varying level of fidelity. We define three levels of representation fidelity for gestures with respect to hand pose and motion. Low-fidelity representations are gesture labels annotated by humans (e.g. "hammering"), while high-fidelity representations are numerical values of hand joint positions or motion trajectories (e.g. trajectory of the hand center in 3D space) that usually are output representations of automatic hand detection systems. Mid-fidelity representations are human textual descriptions of hand shapes or hand motion (e.g. fist moving up and down). We evaluate LLM's ability to reason at different fidelity levels of gesture representation with a focus on textual representations (mid- and low-level fidelity. We initiliazed *GestureInstruct* using low-fidelity gesture class labels, and then populated them with mid-fidelity textual representations by describing the hand shape and motion of the gesture in detail.

**GIRAF can reason about gestures with textual representations of different levels of fidelity.** Fig. 3 shows the results of GIRAF reasoning with both gesture labels (low-fidelity) and gesture descriptions (mid-fidelity). GIRAF is able to achieve 75% success rates on deictic gestures and about 50% success rates on both iconic and semaphoric gesture. Semaphoric gestures are one-shot generalization scenarios (human defines what to do in speech once and the robot needs to learn to do that again next time without speech) so we expect LLM to perform the same with either type of gesture representation. Unexpectedly, the LLM planner completely fails to correctly reason about symbolic gestures when we use gesture descriptions while ChatGPT is able to describe these gestures in our preliminary experiments. Further investigation is needed to understand this behavior.

## 5   Discussion

**Summary.** In this work, we propose the GIRAF framework that incorporates gesture information for multimodal instruction following in human-robot interaction settings. We demonstrate instantiations of GIRAF enable more-favorable and better-performing robots (70% higher success rate) than language-only methods with a user study in two table-top manipulation tasks. We also evaluate GIRAF on the *GestureInstruct* dataset and show that it is able to reason about a diverse set of gestures.

**Limitations and Future Work.** In this work, the instantiation of GIRAF can only handle static gestures. While we show our reasoning module can handle dynamic gestures, we lack a model that can robustly detect dynamic gestures. One can build a specialized dynamic gesture detector through collecting supervised training data or leverage VLMs. While the VLMs we had access to do not take multiple images as input, we expect video-based VLMs can describe human dynamic gestures in the near future. As for the types of gesture descriptions, in our preliminary results we found that GIRAF cannot achieve non-zero performance with high-fidelity numerical representations of gestures (i.e. hand joint positions, 3D motion trajectories). Our conjecture is that existing LLMs lack the level of geometric reasoning required to understand complex numerical representations of gestures. We leave full investigation of this for future work. For similar reasons, GIRAF as a framework still cannot solve tasks that require complex reasoning about the motion of the gestures, such as manipulative gestures (e.g. human user demonstrates a task and says "do this"), which is an interesting open problem for future work. Finally, we focus on hand gestures in this work while human full-body gestures could also be informative and useful for human-robot interactions, which we consider as promising future extensions of GIRAF.

**Acknowledgments**

This work is supported by NSF #2006388, #2132847, #2218760, ONR, AFOSR, and DARPA YFA.

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

# Appendix

## Table of Contents

## A  Gesture Types

We reference the gesture classification in Vuletic et al. [15] for gestures in HRI and adopt the gestures types that are most relevant and merged a few types that are tightly related and makes no difference for detecting purposes. E.g. we merged *metaphoric* gestures with *iconic* gestures as it is defined as iconic gestures for abstract concepts. We summarize the gesture classes in Table 1. Below, we list example gestures of each type in Table 1:

- **Symbolic**: thumbs-up for OK; rubbing index finger and thumb to mean money.

- **Semaphoric**: gestures used in sign language; gestures used for commanding animals.

- **Pictographic Iconic**: gesture a circle to mean a round object.

- **Spatiographic Iconic**: gesture up and down and emphasize the object's location.

- **Kinematographic Iconic**: rolling hand motion to refer to a rolling object.

- **Metaphoric Iconic**: a cutting gesture to indicate a decision has been made.

- **Deictic**: pointing gestures used to indicate an object; palm-open indicating a desire to receive something.

## B  *GestureInstruct* Dataset: Task Scenarios

We generated a total of 36 different gesture-speech instruction-following task scenarios and list them out in Table 2 and Table 3. We do not distinguish the sub-types of gestures under iconic gesture.

Table 1: Classification of Communicative Gestures

| Gesture Type | | Definition | Speech Dependency |
|---|---|---|---|
| Symbolic | | represent an object or concept, have conventional meaning and can be directly translated into words | independent |
| Semaphoric | | used to trigger a predefined action, defined in a formalized dictionary, developed for specific purpose | |
| Iconic | - | represent meaning closely related to the semantic content of the speech, illustrate what is being said | dependent |
| | Pictographic | represent shape | |
| | Spatiographic | represent spatial relation | |
| | Kinematographic | represent action of an object | |
| | Metaphoric | represent abstract concepts | |
| Deictic | | refer to an entity that is being said | |

To call that LLM reasons successfully on a task scenario, it needs to call the correct action primitive with the correct arguments in the generated plan. Take the fist gesture in *semaphoric* category as an example. The goal state is "robot moves to the hand position," so it is considered a successful trial if the LLM generated code looks like this:

```
# Instruction 0: move over here
# Gesture: fist
target_pos = detect_hand_center_pos()
move_gripper_to_pos(target_pos)
```

Since *GestureInstruct* is designed solely for the purpose of evaluating if LLM has zero-shot reasoning capability for these types of gestures, we took a top-down approach of exemplifying gesture types when constructing dataset. However, a bottom-up approach of collecting a suite of gesture-language instructions from real applications would be preferred in practice.

Table 2: *GestureInstruct* Dataset - part 1: Symbolic and Semaphoric Gestures

| Gesture Type | gesture label | gesture description | language instruction | context | intent / goal state |
|---|---|---|---|---|---|
| Symbolic | thumbs up | thumb extends out and points upward while other fingers curl inward | - | robot gripper open and above cup handle | robot picks up the cup |
| | thumbs down | thumb extends out and points down while other fingers curl inward | - | robot gripper open and above cup handle | robot asks human how to make corrections |
| | OK | thumb and index finger forms a circle while others extend out | - | robot gripper open and above coffee pod | robot picks up the coffee pod |
| | stop | an open palm faces outward | - | robot moving towards the person | robot stops |
| | beckoning | an open palm faces inward, and the whole hand moves inward and outward repeatedly | - | robot is far away from the person | robot moves towards person |
| Semaphoric | fist | a closed palm | come over here | - | robot move to hand position |
| | pick up | an open palm first faces upward, and then all fingers curl inward | grasp it | - | robot picks up the coffee pod |
| | release | a closed palm first faces downward, and then all fingers extend out | drop it | - | robot releases the coffee pod |
| | circling horizontally | index finger extends out while others curl inward, and the whole hand moves in a circular motion horizontally | turn around | - | robot turns around |

Table 3: *GestureInstruct* Dataset - part 2: Iconic and Deictic Gestures

| Gesture Type | gesture label | gesture description | language instruction | context | intent / goal state |
|---|---|---|---|---|---|
| Iconic | circle | two hands forms a circle | give me the bowl that shaped like this | there are a square bowl and a round bow | robot hand over the round bowl |
| | measuring with two fingers | thumb and index finger extend out while others curl inward | add this much water | robt is holding a water jug | robot add water in cup with height similar to the size between human's two fingers |
| | pinching | thumb and index finger extend out while others curl inward first, and then thumb and index finger touch each other | do this | robot gripper open and above coffee pod | robot picks up the coffee pod |
| | spreading | thumb and index finger extend out and touch each other while others curl inward first, and then thumb and index finger separate | do this | robot is holding a coffee pod | robot releases the coffee pod |
| | twisting | thumb and index finger extend out while others curl inward, and the whole hand rotates | do this | robot gripper is on a bottle of water | robot twists the bottle cap to open it |
| | pushing | an open palm faces outward, and the whole hand moves outward | do this | robot gripper is on a wooden block | robot push the block towards a specified direction |
| | lifting | an open palm faces upward, and the whole hand moves upward | do this | robot is holding a bottle of water | robot lifts the bottle of water off the ground |
| | squeezing | fingers gently curl inward while thumb is on the other side first, and then all fingers move closer to become a closed palm | do this | robot is holding a lemon and its gripper is above a cup | robot squeezes the lemon |
| | hammering | a closed palm, and the whole hand moves up and down | I am looking for a tool | robot sees hammer, screwdriver, and wirecutter on the table | robot hand over the hammer |
| | cutting | index and middle finger extend out while others curl inward, and index and middle finger repeatedly touch each other and separate | I am looking for a tool | robot sees hammer, screwdriver, and wirecutter on the table | robot hand over the wirecutter |
| | hammering | a closed palm, and the whole hand moves up and down | give me the tool that does this | robot sees hammer, screwdriver, and wirecutter on the table | robot hand over the hammer |
| | cutting | index and middle finger extend out while others curl inward, and index and middle finger repeatedly touch each other and separate | give me the tool that does this | robot sees hammer, screwdriver, and wirecutter on the table | robot hand over the wirecutter |
| | screwing | a closed palm, and the whole hand rotates | give me the tool that does this | robot sees hammer, screwdriver, and wirecutter on the table | robot hand over the screwdriver |
| | circling vertically | index finger extends out while others curl inward, and the whole hand moves in a circular motion vertically | again | robot just drew a square on a piece of paper | robot draws another square |
| | smoking | thumb and other fingers form a curved shape, and the whole hand is near the mouth | bring me this | robot sees various objects on the table | robot hand over a cigarrete |
| | putting on a beanie | two closed palms are on two sides of the head, and both hands moves downward | can you give me this | robot sees beanie and cap on the rack | robot brings a beanie |
| | opening a book | two open palms first face each other, and then they slowly move apart and both face upward | bring this to me | robot sees various objects on the table | robot hand over a book |
| | drinking | fingers gently curl inward while thumb is on the other side, and the whole hand first faces upward and then tilts inward | I need this | robot sees various objects on the kitchen counter | robot hand over a drink |
| | opening a door | fingers gently curl inward while thumb is on the other side, and the whole hand faces outward and rotates | can you help me with this | robot is in a room with a human | robot opens the door |
| Diectic | pointing (object) | index finger extends out while others curl inward | pick up this water jug | robot gripper is empty | robot picks up the water jug |
| | pointing (location) | index finger extends out while others curl inward | put it over here | robot is holding a water jug | robot place down the water jug at the pointed location |
| | pointing (driection) | index finger extends out while others curl inward | move a little bit this way | robot is holding a water jug | robot moves in pointed direction |
| | pointing (multiple objects) | index finger extends out while others curl inward | throw away this, this, and this | robot see multiple objects on a dirty table | robot throw away the referred items into trash can |
| | handover | an open palm faces upward | place it over here | robot has a tool in hand | robot hand over tool |
| | touching an object | an open palm faces downward, and the whole hand is on an object | pick up this | robot gripper is empty | robot picks up the touched object |
| | drawing | index finger extends out while others curl inward, and the whole hand moves | draw this | robot is holding a pencil on a piece of paper | robot draws a similar shape on the paper |
| | circling (dynamic pointing) | index finger extends out while others curl inward, and the whole hand moves | throw all the trash into the trash can | robot gripper is embpy | robot throw all the referred trash into the trash can |

## C    User Study

### C.1    Definition of Success

As shown in Fig. 2, we consider two types of success rates in this user study:

1. Success rate:

   This refers to the planning success rate. We count a test successful if the robot finds the correct object (e.g. drawer, tool) that the user wants within 3 trials. This measures how good the task planner and referent detection heuristic is, which is the focus of this work.

2. Success rate (execution):

   This refers to the full task success rate. We count a test successful if the robot can complete the task (e.g. open the correct drawer, fetch the correct tool) within 3 trials. This measures the performance of the whole system, including the robot execution.

### C.2    Instructions

We provide the instructions we present to the user below:

In this study you will interact with a robot arm to complete 2 different table-top manipulation tasks and each in 2 different scenarios. The study will take approximately 45 minutes to complete. Please open the survey and fill out the first section before continuing.

**Task: Open Cabinet Drawers** In this task, you will ask the robot to open a few drawers of the cabinet for you. Pick 3 drawers (within the highlighted area) you want to ask the robot to open and mark them in this image (mark the order you want to have them opened): <image> You will repeat this task in two different scenarios:

- Scenario M: You use language to describe which drawer to open.
- Scenario N: You use both language and pointing gestures for specifying the target drawer to open.

When you are ready to initiate a task, step once on the foot pedal and issue an instruction for the robot. The robot will confirm your instruction by repeating what it heard. You have 3 trials for each task (speech detection failure does not count as a trial). Please fill out the corresponding survey questions after each scenario.

**Task: Fetch Tools** In this task, you will ask the robot to fetch a few tools for you. Pick 3 tools (randomly) for the robot to fetch and mark them with the order you'd like to have them in this picture: <image> You will repeat this task in two different scenarios:

- Scenario M: You use language to describe which tool to fetch.
- Scenario N: You use both language and pointing gestures for specifying the target tool to fetch.

When you are ready to initiate a task, step once on the foot pedal and issue an instruction for the robot. The robot will confirm your instruction by repeating what it heard. The robot will try to hand the tool over to you. You have 3 trials for each task (speech detection failure does not count as a trial). Please fill out the corresponding survey questions after each scenario.

### C.3    Survey Questions

For each task, the user fill out the following survey for each scenario by rating how much the agree with each statement:

- The robot is able to understand my instruction.
- The robot is able to identify the correct item I wanted.
- The robot is able to successfully execute the task.
- It is easy to communicate with the robot what I wanted.
- I feel frustrated after interactions with the robot.

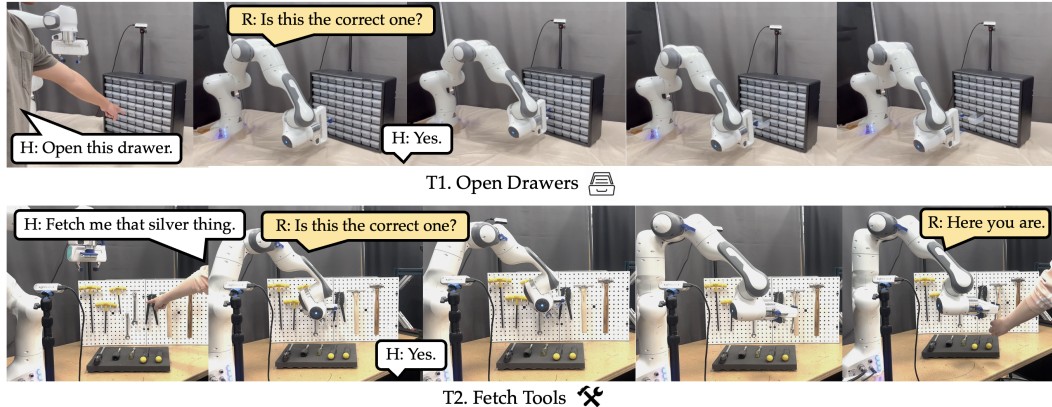

T1. Open Drawers

T2. Fetch Tools

Figure 5: **User Study Task Rollouts**

- I trust the robot for completing tasks I specified for it.
- I would use this robot again in the future for similar tasks.

### C.4 Rollouts

Example trajectories of the two user study task are shown in Fig. 5. We record all the task specification language and RGB-D images of the scene from 3 calibrated camera views and will release this data on our website as well.

## D  Motion Primitives

We adopt the method from Sundaresan et al. [30] and implement a library of parameterized motion primitives for the tasks in our experiments. All of these motion primitives are parameterized with a 3D position in the robot's reference frame. For example, the primitive for picking up an object takes the center point of the object as parameter and the placing primitive takes the target location as parameter. The depth cameras are fixed and calibrated so that the detected object positions can be mapped to the robot's reference frame. For simple pick-and-place motions, we hand code motion primitives that command the gripper to move to the input target position with a fixed (upright) orientation. For complex manipulation motions such as opening the drawer and picking up non-concave objects (tools), we use waypoint-based motion primitive models. To train each motion primitive model, we collected supervised gripper pose data conditioned on a pointcloud input and a 3D point on the object de-projected from a 2D point annotated by the user. At test time, the detected object centers output from the scene descriptor will be used as input to these motion primitive models.

## E  Prompting LLM

We prompt LLM for tasking planning, including reasoning about what to perceive. We also implement a language program to filter target objects. For our basline implementation, we also prompt the LLM to reason about spatial relationships between objects with examples. Full prompts used in our experiments can be found on our project website: `tinyurl.com/giraf23`

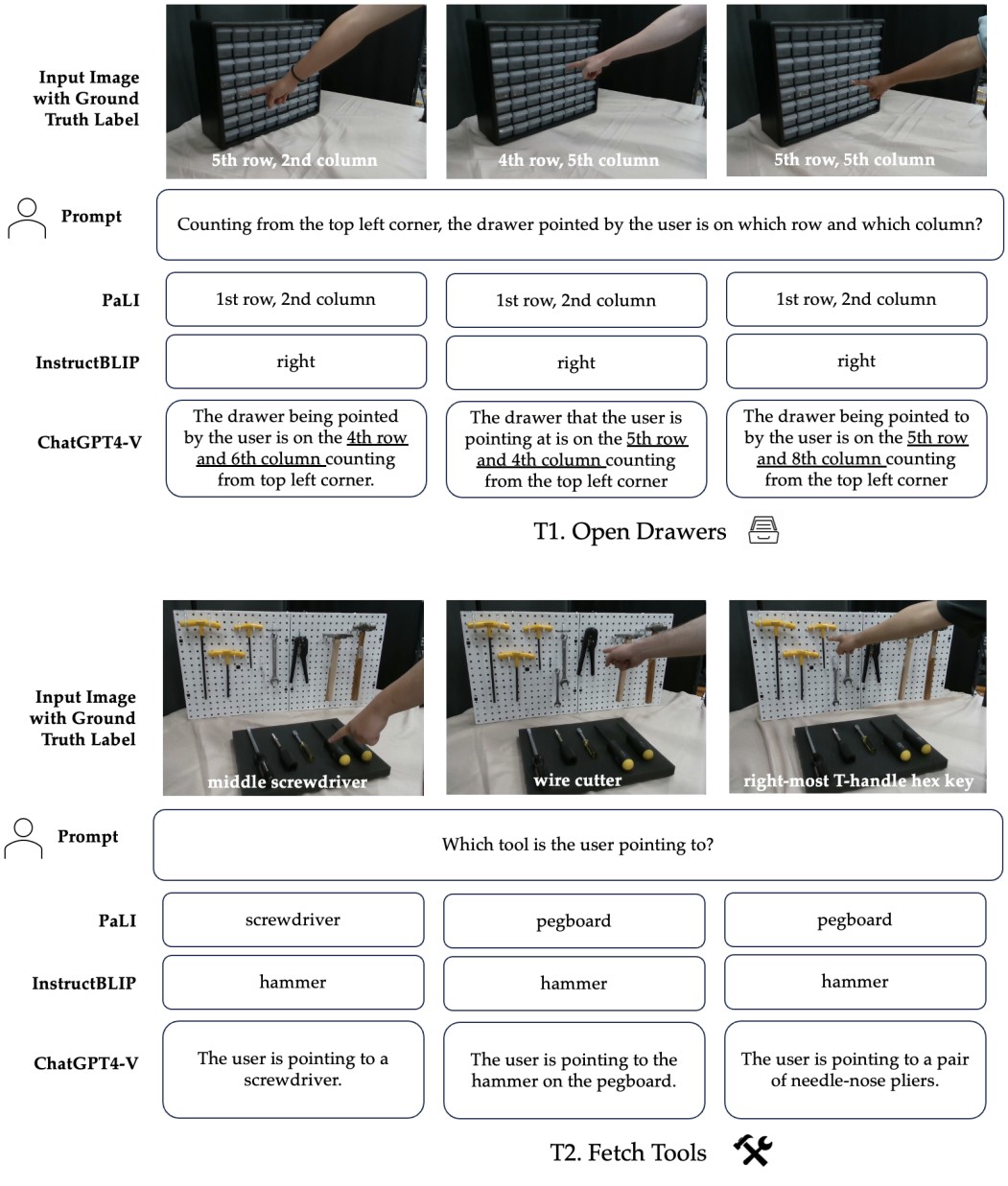

Figure 6: **VLM as referent detector.** Examples of running referent detection with PaLI [31], InstructBLIP [32], and chatGPT4-V demonstrate that existing VLMs still struggle with fine-grained geometric reasoning.

# F VLM for Grounding Gestures

We investigate VLM's capability of both 1) grounding gesture class and 2) detecting the referent of a pointing gesture. We test 3 state-of-the-art models: PaLI [31], InstructBLIP [32], and chatGPT4-V.

VLMs take the full scene image as input and therefore have more context to reason about the gesture type than an end-to-end gesture classifier trained with only hand keypoints features and therefore has the potential to replace data-driven gesture classifiers. We prompt InstructBLIP [32] to classify static gestures in the *GestureInstruct* dataset and it can achieve around 80% zero-shot accuracy.

However, VLMs are not good at predicting referent of the pointing gesture, as shown in Fig. 6: none of the three models can output the correct drawer instance in the open drawers task as it requires complicated geometric reasoning: chatGPT4-V struggles to identify the exact location of the drawer, PaLI gets the answer format correct but outputs the same answer for different input images, while InstructBLIP does not even output reasonable answers); all three models can predict reasonable output in the fetch tools task (chatGPT4-V and PaLI generating plausible responses in all cases) but lack the sensitivity for precise referent detection.

## G  Realistic Use Cases

While we designed the tasks for user study to highlight the benefits of combining gesture understanding when following language instructions, we believe the problems GIRAF addresses broadly exist in many realistic scenarios. For example, the Fetch-Tool task represents tasks where the human user does not know the name of the target item or the robot cannot detect the objects as the human recognizes them. From the user's end, this is common when English is not their native language. From the robot's end, this may very well happen when it encounters rare objects or objects with unusual appearance. If the user could realize the robot doesn't understand what they are referring to, they would want to point at the target instead.

## H  Demos of Additional Use Cases

We show additional use cases of GIRAF including (1) using multiple gestures in a single instruction: opening multiple drawers (Fig. 7), and picking and placing a tool (Fig. 8); (2) specifying grasp point on an object (Fig. 9); and (3) instructing the robot to perform a long-horizon sorting task (Fig. 10).

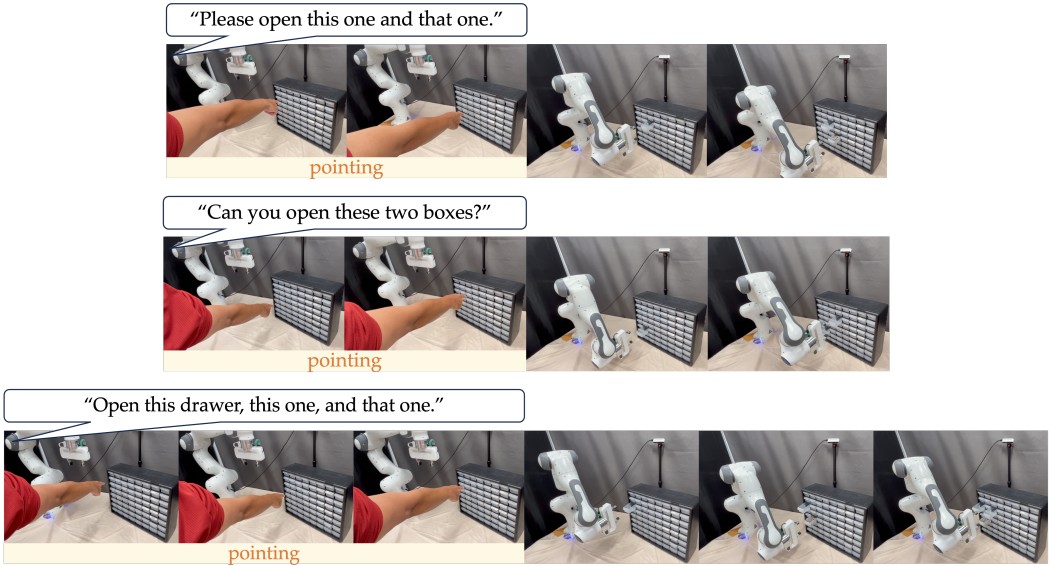

Figure 7: **Open multiple drawers at once.**

## I  Failure Modes

We summarize error modes of each module in the system below. We generally take two measures to account for different error modes. The first one is to make the system as transparent as possible so we can catch the error before it propagates to the next level (e.g. robot will announce the action it plans to take and confirm the target with the human user before it takes any action). The second measure is to catch exceptions in the code and detect simple failures (e.g. syntactic errors and grasp failures). For code error, the robot will announce its failure and ask the user to try again, for grasping

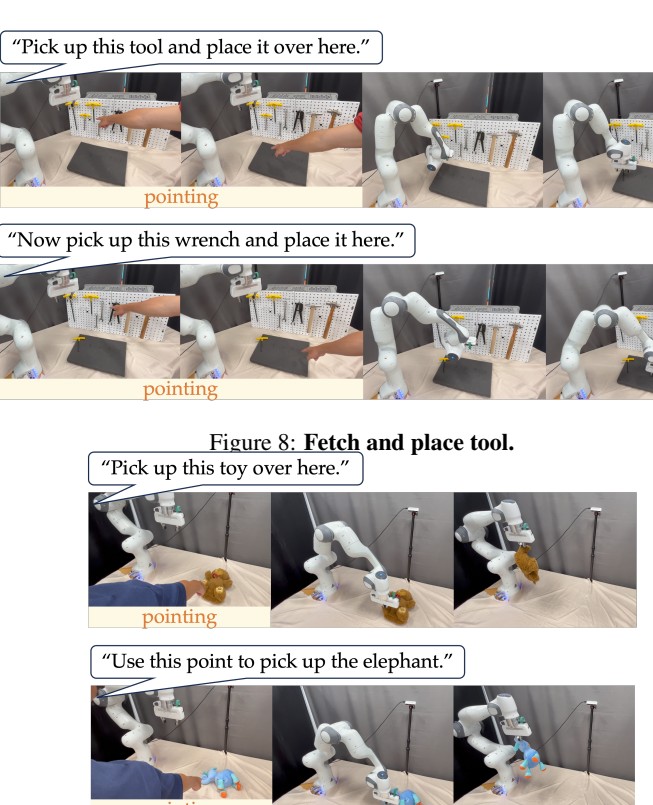

Figure 8: **Fetch** and place tool.

Figure 9: **Pick up object from a specific point.**

failure the robot will adjust its motion primitive parameters and try again. While we do not directly tackle low-level action primitive failures, our system allows the user to fix low-level execution errors by incorporating directional pointing gestures. In one of our long-horizon tasks (Fig.4 in the paper), we show that the robot missed the grasping point of the mug handle by a bit and the user can point in the direction they want the robot to move in order to fix it.

1. Human description:
    (a) Speech (Azure API [27]):
        • Misrecognized words:
          This happens more often for non-native English speakers than the native ones as we observed in our user study. On average, this happens to native speakers 0.27 times per instruction and non-native speakers 0.46 times per instruction. When the user's instruction is misrecognized, we ask the user to give the instruction again, so this error would not propagate into the system.
    (b) Gesture:
        On average, gesture is not detected 0.07 times per instruction, and we further discuss different reasons of failure below.
        i. Hand detector (MediaPipe [28]):
            • Hand not detected:
              This happens more often when the image is taken from the side of the hand instead of the back or front of the hand. When this happens, we ask the user to give the instruction again, so this error would not propagate into the system.
            • Incorrect hand keypoints:
              For example, it might label the joints of the thumb as joints of the index finger. This happens more often when some hand joints are occluded in the image, such as doing a fist gesture, and if this happens, we would still feed those keypoints

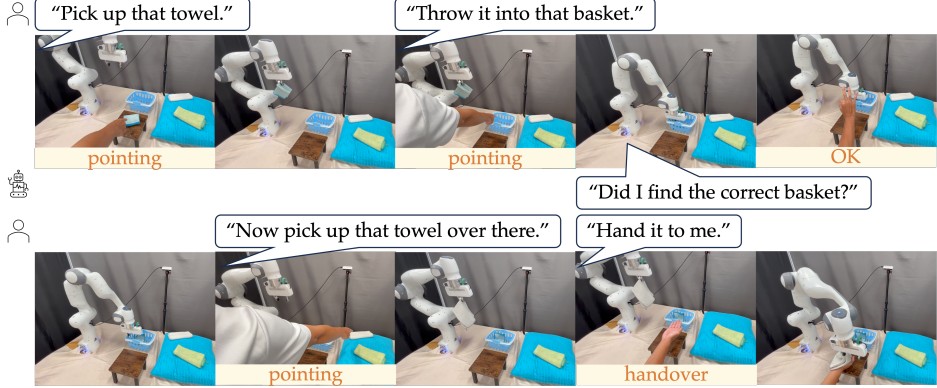

Figure 10: **Long horizon interaction with multiple gestures.**

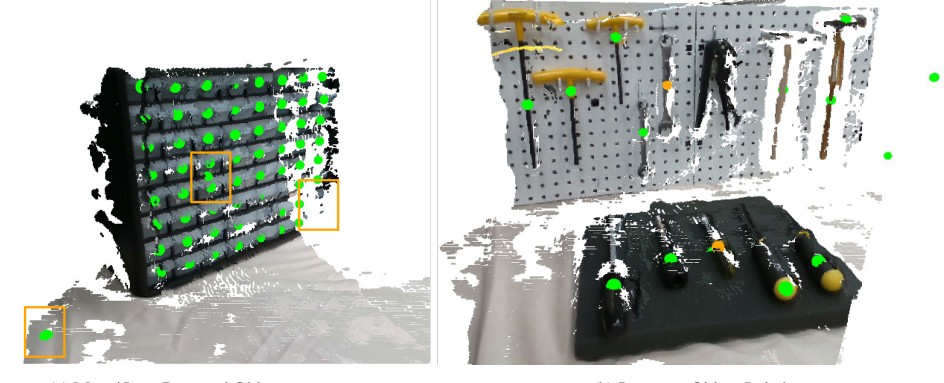

(a) More/ Less Detected Objects.                    (b) Incorrect Object Labels.

Figure 11: **Scene Descriptor Failure Modes.**

into the gesture classifier because we don't have the groundtruth to verify whether the keypoints are correctly labeled. This kind of error might hurt the accuracy of the gesture classifier and referent detection module because both of them directly take hand keypoints as input instead of raw images.

ii. Gesture classifier
- Predict as unknown gesture:
  In this case, we would treat it as no gesture detected and ask the user to give the instruction again, so this error would not propagate.
- Predict as another gesture:
  For example, the gesture classifier might predict a pointing gesture as a fist gesture. In this case, we would still feed the predicted gesture to the reasoning module as we don't have the groundtruth. This kind of error can hurt the performance of the LLM task planning module. For example, when the user says "move over here" while the gesture classifier predicts a pointing gesture as a fist gesture, LLM would tell the robot to move to the hand center instead of moving to the pointing location.

2. Scene description:
- More/ less detected objects:
  As shown in the orange regions in Fig. 11a, the scene descriptor might predict more or less drawers than the actual number, but this is generally not problematic in our experiments because users don't usually point at those misclassified objects.
- Incorrect object labels:
  Another case of failure is when the scene descriptor gives objects wrong labels. This would hurt the performance of the referent detection module as it would apply a

semantic filter to just consider a subset of objects given object labels and the user's instruction. For example, when the user says "pick up the leftmost screwdriver," the referent detection module would only consider the two orange points in  that are labeled as screwdrivers to be candidates that the user might point at. Therefore, even if the user is pointing at the leftmost screwdriver, our system would think the middle one is the one being pointed at.

3. LLM task planning:

   - Bugs (semantic and logic errors in generated code):
     Bugs happen mainly because of its confusion about Python list and Numpy array. As a result, the generated code might try to call a Numpy array function on a Python list, resulting in an inexecutable piece of code. Bugs would also be generated when the language instruction is out-of-distribution. One example during our user study is when the user said "on the third row take the third drawer." Since it differs from the sentences we put in the prompts too much, the generated code tries to feed more arguments into the action primitive. Specifically, the action primitive for opening a drawer takes one argument of the drawer's 3D position, but in this example, the generated code wants to feed the function with an additional 3D position corresponding to the third row. On average, bugs happen 0.02 times per instruction.

   - Incorrect gesture interpretation:
     LLM might also fail to understand the meaning of some gestures. For example, LLM fails to call the correct function `navigate_following_user()` given the gesture "beckoning."

4. Referent detection:

   On average, referent detection fails 0.2 times per instruction. Besides being affected by accumulated errors from other components, we also observe another case of failure that is related to user's behavior.

   - User's different behavior (unreliable pointing):
     We observed one interesting case where some users tend to use gestures as a vague indicator of their attention and still rely on language to provide the accurate description. For example, one user gave the instruction "open the drawer on the second row and second column" while vaguely pointing toward that direction. Therefore, for this kind of users, our heuristic fails more often because their pointing gesture might not actually point at the object, but we also found out these cases happen more at the start of their user studies as they were able to adapt to our system by pointing more accurately through a few trials.

## J   Influence of Distance on Referent Detection

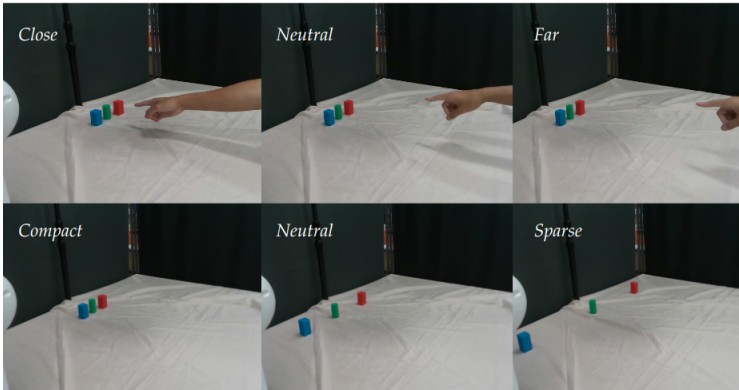

Figure 12: **Setup for Testing Referent Detection Accuracy.**

Table 4: Referent Detection Accuracy under Different Setups.

|         | Close | Neutral | Far |
|---------|-------|---------|-----|
| Compact | 2/3   | 2/3     | 2/3 |
| Neutral | 2/3   | 3/3     | 3/3 |
| Sparse  | 3/3   | 3/3     | 3/3 |

To further test how our referent detection heuristic works as the user standing further away, we ran a small experiment varying the user's distance to the objects and how close the objects are. The setup is shown in Fig. 12. As shown in Table 4, our heuristic can still work even when the user points from further away, and as the distance between objects becomes larger, the accuracy of our heuristic improves because it is less likely to be affected by noise in the detected hand keypoints.

