# OpenReview forum: "Gesture-Informed Robot Assistance via Foundation Models"
_robot-learning.org/CoRL/2023/Conference — CoRL 2023 Poster_

### Official Review · Reviewer_beCJ · 2023-07-19

**Confidence:** 4
**Originality:** Very Good
**Technical Quality:** Excellent
**Clarity Of Presentation:** Excellent
**Impact:** 4

**Recommendation:**

Strong Accept: I recommend accepting the paper and will argue for my recommendation even if other reviewers hold a different opinion.

**Review:**

Overall, I thought this paper was very well-written and enjoyable to read! The framework is sound, novel and clever; the experiments are thorough and compelling.

I have 2 comments that I’d like to see addressed.

First, a strong motivation for the paper is that gestures can be ambiguous (e.g. pointing at a cup could mean “pick up the cup” or “pour in this cup”), yet looking at Fig. 3 every gesture has only one corresponding language utterance. I would’ve liked to see language resolving gesture intent ambiguity, yet as far as I can tell that isn’t explored. Essentially, the user study shows that gestures are helpful for disambiguating language, but the intro also motivates the other direction: how language can be useful to disambiguate gesture intent. Have you tried an experiment like this, where you measure success rate for disambiguating between multiple meanings of the same gesture?

My second comment is that I’d like to see more discussion on the types of situations where the human knows how to use an object yet can’t name it. In the “open this drawer” it is clear why using gestures is important: the human clearly could say “open the 3rd drawer on the 2nd row”, but that is cumbersome and gesturing helps with that. But in a situation like “give me this tool”, when would a human not know how to name the tool yet know how to use it? What prevents the human from saying “give me the screwdriver”? I think it’s really important for the applicability of the method to have some discussion on when these kinds of situations emerge, otherwise the reader might be left feeling like this problem is a little bit manufactured.

Small comments:
- Abstract typo: “deictib” should be “deictic”
- Intro typo: “a pointing” should just be “pointing”
- Fig. 3 caption “gestuers” → gestures
- Discussion: “langauge” → language


**Quality Of The Limitations Section:**

Limitations are addressed clearly

**Questions For Rebuttal:**

For the rebuttal, please discuss the gesture ambiguity issue. Can you either have a small experiment on that or discuss why the method should or should not work on it? Please also add a discussion on the situations where the method applies to demonstrate that this is not a niche problem. Lastly, please fix the grammatical errors; the paper is rife with them.

**Robotics Focus:**

Sufficient demonstration on hardware

**Summary Of Paper:**

This paper’s goal is to enable robots to recognize gestures and contextualize them using real time language instructions from humans. The challenge here is that the intent behind a gesture strongly depends on the context (pointing at a cup can mean pour into the cup or pick up the cup, but you don’t know which one the human wants without more information than just the gesture). Similarly, language on its own can be ambiguous without gestures ("open this drawer" is very unclear). They propose a framework that leverages LLMs to contextualize the meaning of human gestures and infer human intent from them. The idea is to have a model for identifying the gesture, then prompt an LLM for reasoning about the gesture-language pair and generate a robot policy from that. The paper demonstrates that the proposed method, GIRAF, is preferred by users in a user study when compared to a language-only baseline, and also curates and publishes a dataset of gestures that can be useful for future research.

**Summary Of Recommendation:**

I really like this paper and I think it would be a wonderful contribution to CoRL! I think the ideas presented are novel, the multimodal framework could potentially be very useful in the embodied robotics community, and the gestures data set would be helpful for future researchers. The paper is somewhere between a weak and a strong accept for me, mostly because of the 2 issues above I would like to see addressed.

---

> ### Author Response · Authors · 2023-08-14
> **Rebuttal Period Closing: additional comments?**
>
> We'd like to remind the reviewer that the rebuttal period is ending. We truly appreciate your valuable feedback on our paper. As the rebuttal period concludes tomorrow, we assure you that we have included all the discussion points brought up in the review in our revision. If there are any additional points you'd like us to discuss or consider, please let us know! Your insights have been invaluable, and we're grateful for your contribution to our work.

---

### Official Review · Reviewer_zpDJ · 2023-07-19

**Confidence:** 4
**Originality:** Good
**Technical Quality:** Good
**Clarity Of Presentation:** Very Good
**Impact:** 3

**Recommendation:**

Weak Accept: I recommend accepting the paper, but will not argue for my recommendation if the majority of other reviewers have a different opinion.

**Review:**

Strengths:
1. The idea is technically solid and straightforward.

2. The writing is clear and the videos are helpful.

3. Analyses and discussions are quite informative.

Weaknesses:
1. The method involves combining multiple existing models. It would be good to estimate errors from each model and how errors can be accumulated in difficult cases. E.g., what will happen if perception models fail?

2. It would be good to discuss what type of tasks could be solved by this system with or without gestures. I imagine that some tasks can be directly solved by instruction following (language only) without gestures, while others would have to rely on accurate gesture recognition. And there could be tasks that simple gestures are not sufficient.

3. LLMs-based planning does not inherently have a safety guarantee. How would you address failures and potentially unsafe plans generated by LLMs?

**Quality Of The Limitations Section:**

Additional details required

**Questions For Rebuttal:**

Please address the two concerns in the review.

**Robotics Focus:**

Sufficient demonstration on hardware

**Summary Of Paper:**

This paper presents an approach to prompt LLMs for robot planning conditioned on language instructions and gestures. It first uses off-the-shelf perception models to parse the scene and recognize human gestures, and then prompts LLMs with both the scene and gesture descriptors (as code/templates) as well as the language instructions to generate task plans. It then calls low-level controllers to execute the plans. Evaluation on a real robot shows promising results.

**Summary Of Recommendation:**

This paper presents an interesting idea to use LLM for gesture & language-guided robot planning. There are concerns about safety and robustness to errors made by individual modules. My final rating will depend on how authors address these concerns.

---

> ### Author Response · Authors · 2023-08-14
> **Rebuttal Period Closing: additional comments?**
>
> We'd like to remind the reviewer that the rebuttal period is ending. We truly appreciate your valuable feedback on our paper. As the rebuttal period concludes tomorrow, we assure you that we have included all the discussion points brought up in the review in our revision. If there are any additional points you'd like us to discuss or consider, please let us know! Your insights have been invaluable, and we're grateful for your contribution to our work.

---

### Official Review · Reviewer_h9cL · 2023-07-20

**Confidence:** 4
**Originality:** Good
**Technical Quality:** Good
**Clarity Of Presentation:** Good
**Impact:** 3

**Recommendation:**

Weak Reject: I recommend rejecting the paper, but will not argue for my recommendation if the majority of other reviewers have a different opinion.

**Review:**

**Strengths**

- The paper proposes a "relatively" complete system for identifying gestures and human language, and feeding this information into the LLM planner and some robot motion execution primitives.
- A user study is included in the paper, making the evaluation more objective.
- The whole system is pretty straightforward.

**Weaknesses**

- Although the paper reports quantitative data, there is not enough clear context provided for each number. For instance, what does the success rate represent? How many total tests were conducted? This lack of clarity applies to many parts of the paper, such as Figure 2, 3, or the success rate on the GestureInstruct dataset.
- From a high-level perspective, humans typically use gestures when they are at a distance from objects. However, in the examples shown in the paper (drawer opening, tool fetching), the human operator is very close to the objects, to the point that there isn't really a need for robots to do the task. Humans can do them themselves. Therefore, these tasks appear somewhat contrived. To make the approach more compelling, it would be beneficial to show examples where human subjects stand far away from the object, and the robot can still identify the human gesture and where the fingers are pointing at, and complete the task. It might be the case that when humans are far away from the object, it would become more challenging to correctly identify the gesture and where the human is pointing at, and hence, the proposed pipeline might not work well.
- The system has many components, and it is likely that each component can have its own failure modes. For instance, when does the heuristic-based referent searching fail? Unfortunately, there is limited discussion on these failure modes at present.
- It seems that there are no physical experiments for gestures other than "pointing"?

**Quality Of The Limitations Section:**

Limitations are not well addressed

**Questions For Rebuttal:**

* Regarding the drawer opening task, does the depth camera perform well for these small transparent drawers? I assume that the estimated positions of the drawers will have higher errors. If yes, how did the authors obtain good estimates?

**Robotics Focus:**

Sufficient demonstration on hardware

**Summary Of Paper:**

This paper presents a system investigating hand gestures for enhanced human-robot interaction by incorporating off-the-shelf components from prior work. The core contribution appears to be demonstrating the value of gesture information for HRI through the proposed system, rather than introducing novel technical methods.

**Summary Of Recommendation:**

The paper investigates an interesting use of gestures for human-robot interaction. However, the experiments conducted so far have been shown in contrived setups. It remains unclear how well the system will function when human subjects are far away from the object, where gestures are more important in practice. Additionally, the paper only has limited real-world experiments demonstrating how the system works with other gestures, and the limitations are not thoroughly discussed.

---

> ### Author Response · Authors · 2023-08-14
> **Rebuttal Period Closing: have we addressed your concerns?**
>
> We'd like to remind the reviewer that the rebuttal period is ending. We truly appreciate your valuable feedback on our paper. As the rebuttal period concludes tomorrow, we hope that we have clarified any confusion and addressed your concerns in our response. If there are any additional points you'd like us to discuss or consider, please let us know.
>
> We kindly request your consideration for a potential score adjustment based on our responses. Your insights have been invaluable, and we're grateful for your contribution to our work!

---

### Official Review · Reviewer_i78B · 2023-07-23

**Confidence:** 4
**Originality:** Good
**Technical Quality:** Good
**Clarity Of Presentation:** Good
**Impact:** 3

**Recommendation:**

Weak Accept: I recommend accepting the paper, but will not argue for my recommendation if the majority of other reviewers have a different opinion.

**Review:**

Strengths
- Incorporating gestures into instruction following tasks is very useful for designing effective human-robot interaction.
- The robot experiment conducted in this paper demonstrates that users consider using gestures as an easy and preferred way to communicate.

Weaknesses
- It is unclear what the role of LLMs is from the description of the paper. It seems to me that the LLMs can interpret the same pointing gesture differently based on the language instruction and the history. However, there is no experiment to evaluate this context-dependent reasoning which left a question that if we need an LLM here or if it is just to help to parse the instructions.
- The perception API identifies the object/location the user is pointing to. Then a simple approach can be just extracting the action (i.e. verb) referred to in the instruction and then completing the action with the object intersecting with the pointing ray. How does the proposed LLM-based approach compare to a baseline like this or other prior works that also consider pointing gestures?
- The evaluation using GestureInstruct only shows that LLMs can recognize gesture descriptions. However, it doesn’t mean that the classified gestures can be integrated into task planning as the pointing gesture. It will be helpful if the paper can demonstrate how an LLM can take diverse gestures into consideration for task plans.


**Quality Of The Limitations Section:**

Limitations are addressed clearly

**Questions For Rebuttal:**

As discussed in the weaknesses, the paper didn’t describe the role of LLMs and have comparisons with baselines (that also consider gestures) to demonstrate the effectiveness to include LLMs in the instruction+gesture reasoning. It is hard to understand how LLMs contribute to the task in addition to gesture classification and task planning as in CaP.

**Robotics Focus:**

Sufficient demonstration on hardware

**Summary Of Paper:**

This paper proposes GIRAF, an LLM-based framework to include gesture information in instruction following. GIRAF first uses several descriptors to describe scenes (objects and their 3D positions) as well as human speech and gestures. It then uses an LLM-based planner to generate the task plan by prompting with the instruction and the gesture description. To execute the plan, the referred object is identified by intersecting the pointing ray with the candidate objects. The experiment on a real robot demonstrates that GIRAF provides a more effective communication interface and is preferred by humans. To evaluate if the LLMs can understand diverse gestures described in different fidelity, this paper additionally proposes a GestureInstruct dataset for evaluation.

**Summary Of Recommendation:**

It is important to include gestures in the communication modality as it has been demonstrated as effective by user studies. However, the evaluation in its current form cannot justify the benefit of using LLMs in gesture reasoning.

-----
Post-rebuttal update:
I would like to thank the authors for updating the paper to address my concerns. I agree with the authors that the LLM provides context for disambiguating the gesture but I didn't see an experiment that tests how well the LLM helps in disambiguation, for example, just extracting the verb from language and combining it with an object pointed by the gesture can be a good baseline for comparison. In general, I think this is an initial study and a good resource for the community, but I would encourage the authors to include a discussion about the provided information from language (as in the table presented in the rebuttal) in the paper and evaluation on how well LLMs help disambiguate. I'm updating my score accordingly.

---

> ### Author Response · Authors · 2023-08-14
> **Rebuttal Period Closing: have we addressed your concerns?**
>
> We'd like to remind the reviewer that the rebuttal period is ending. We truly appreciate your valuable feedback on our paper. As the rebuttal period concludes tomorrow, we hope that we have addressed your concerns in our response. If there are any additional points you'd like us to discuss or consider, please let us know. We kindly request your consideration for a potential score adjustment based on our responses. Your insights have been invaluable, and we're grateful for your contribution to our work!

---

### Author Response · Authors · 2023-08-12
**Shared Response to Reviewers**

We thank the reviewers for their detailed comments and constructive feedback! We are delighted to see reviewers agree that incorporating  gestures is important for human-robot interaction and glad a majority of reviewers find our framework to be sound and novel, and our experiments to be thorough and compelling.

We acknowledge the **common concerns** raised by the reviewers and summarize them below:
- User study tasks are contrived; lack in-depth discussion of realistic use cases of gestures
- Role of language/LLM is unclear; lack comparison with rule-based baseline
- Lack discussion of failure modes for different components of the system
- Need more discussion of limitations for both the framework and implementation

We present more discussion on the use cases, limitations, and failure modes of GIRAF in our response and update our paper/appendix to include these discussions. The updated paper is attached to individual responses, and the updated parts are marked in orange.

We have also run the following experiments to address concerns about scenarios other than simple pointing and our referent detection heuristic:
* More use cases of (diverse) gestures:
  * Referring to multiple drawers at once e.g. “open this one and that one”
  * Tool placement e.g. “pick up this tool and place it over here”
  * Semantic grasping e.g. “pick up this toy over here”
  * Laundry sorting “pick up that towel”, “throw it into that basket”, OK sign for action approval, and ”hand it over”
  * (Videos of these experiments are on our [project website](https://tinyurl.com/giraf23) and visuals are included in Appendix.H of the updated paper.)
* Referent detection with varying distances between hand and objects and among objects

**Changelist of the updated paper**
- Extended Section 5. Limitations and Future Work
- Added details about evaluation with GestureInstruct in Appendix.B
- Added discussion of the limitation of the dataset in Appendix.B
- Added details about user study evaluation success criteria in Appendix.C.1
- Added discussion of realistic use cases in Appendix.G
- Added visuals of additional experiments of GIRAF in Appendix.H
- Added discussion of failure modes of different components in Appendix.I
- Added results of evaluating the influence of distance on referent detection in Appendix.J

We next provide detailed response to common concerns among the reviewers in this thread.

---

> ### Author Response · Authors · 2023-08-12
> **User study tasks are contrived; lack in-depth discussion of realistic use cases of gestures  (Reviewer beCJ + Reviewer h9cL)**
>
> Multiple reviewers pointed out that the tasks in our experiments seem contrived and the use case of GIRAF can be limited. To this end, we’ve added new experiments (Appendix.H of the updated paper) in hope to show the broad application of our system. We in fact designed the experiments to highlight the benefits of combining gesture understanding when following language instructions, but we believe the problems GIRAF addresses broadly exist in many realistic scenarios. For example, the Fetch-Tool task represents tasks where the human user does not know the name of the target item or the robot cannot detect the objects as the human recognizes them. From the user’s end, this is common when English is not their native language. From the robot’s end, this may very well happen when it encounters rare objects or objects with unusual appearance. If the user could realize the robot doesn’t understand what they are referring to, they would want to point at the target instead.
> Reviewer h9cL also raises the question why users would need the robot if they can just point at the object up close. We agree that in our experiments the human and robot are in close proximity to each other and that can seem contrived but even in close proximity settings  there exist scenarios when the user does not want to directly touch the object they want the robot to handle. For example, if the object is extremely hot or cold, or the object surface is dirty or poisonous. Another concrete example of gestures happening in close proximity is when ordering a sandwich at a loud Subway store, the human is constrained to reach the ingredients themselves even though they’re in close proximity to the object.
> We would also like to clarify that the main factor limiting users from standing far away is the mounting location of our depth cameras, which could be easily improved if we have a larger robot workspace and better camera setup. To further test how our referent detection heuristic works as the user standing further away, we ran an experiment varying the user's distance to the objects and how close the objects are, and measured the referent detection accuracy (results are shown in Appendix.J of the updated paper). The results show that our system can still identify the correct target when the user is pointing from a relatively far distance, but we do observe that when the user’s hand is further away and appears at the edge of the image, our hand detection module (MediaPipe) fails to detect the hand more often, resulting in more retries.

---

> ### Author Response · Authors · 2023-08-12
> **Role of language/LLM is unclear; lack comparison with rule-based baseline (Reviewer beCJ + Reviewer i78B)**
>
> Reviewers beCJ and i78B request more discussion on the role of language and language model.
> We mainly consider language instruction as the context for LLM to infer what additional information should be extracted from the gestures, which is important when there are multiple interpretations of the same gesture. For example, when the user says “pick up that cup” with a pointing gesture, the LLM can infer it needs to parse a referred object ‘cup’ from the gesture. If the user says “move in this direction” instead, the LLM then needs to reason it should parse the direction of the pointing gesture. We list out gestures that have multiple interpretations in the GestureInstruct below. We summarize the information that can be provided by these gestures in the rightmost column. We found that LLM is able to reason correctly for all these cases.
> | Gesture     | Context                                   | Provided Information |
> | ----------- | ----------------------------------------- | -------------------- |
> | pointing    | User says “pick up the water jug”         | object               |
> | pointing    | User says “place it over here”            | location             |
> | pointing    | User says “move a little bit this way”    | direction            |
> | OK          | Robot confirms its action with user       | approval             |
> | OK          | Robot asks how many apples the user wants | number               |
> | thumbs up   | Robot confirms its action with user       | approval             |
> | thumbs up   | User says “move a little bit this way”    | direction            |
> | thumbs down | Robot confirms its action with user       | disapproval          |
> | thumbs down | User says “move a little bit this way”    | direction            |
>
> In the above examples language helps disambiguate gestures and it may seem a rule-based system would also work. There also exist cases where gestures inform how information should be extracted. For example, when the user says “place it here” with a pointing gesture, the information it needs to parse (location on the table) is different from when they use an open-palm (handover) gesture (hand location). In our evaluation, although the LLM planner never sees a handover gesture in its prompt, it is still able to call detect_hand_center_pos() instead of detect_referred_pos() to extract positional information from that gesture, which is beyond the capabilities of a rule-based system without hard-coding this corner case.
> We list out more examples of the same kind below.
> | Context                                      | Gesture   | Extraction Function                          |
> | -------------------------------------------- | --------- | -------------------------------------------- |
> | “place it over here”               | pointing  | detect_referred_pos()                        |
> | “place it over here”               | handover  | detect_hand_center_pos()                     |
> | “pick up this tool”                | pointing  | detect_referred_obj_pos(“tool”)              |
> | “pick up this tool”                | hammering | detect_related_obj_pos(“tool”)               |
> | “move in this direction for a bit” | pointing  | detect_index_finger_referred_direction_vec() |
> | “move in this direction for a bit” | thumbs up | detect_thumb_referred_direction_vec()        |
>
>
> While we agree with Reviewer i78B that it would be great to have a comparison with some rule-based systems, we haven’t found a way to make the comparison informative. Given a specific rule-based system, we can always find a new gesture that it cannot handle. At the same time, to the best of our knowledge, we are the first to look at multimodal (gesture + language) human-robot interaction beyond deictic gestures, so we cannot find a suitable prior work to compare with. Therefore, we only evaluate how our LLM-based task planner performs on GestureInstruct, which are mostly gestures never seen in the prompt (except for pointing and semaphoric gestures).
> We are happy to run comparisons if reviewers could provide suggestions on how to make them informative and pointers to rule-based baselines.

---

> ### Author Response · Authors · 2023-08-12
> **Failure Modes of GIRAF (Reviewer zpDJ + Reviewer h9cL)**
>
> Reviewer zpDJ and Reviewer h9cL requested further details of failure modes of different modules in our system. We agree this is an important topic and discuss the failure modes of each module and how they affect the overall system performance in detail in the appendix of the paper (section I).
> To summarize, every module in the system has error modes that influence the performance of the overall system, e.g. the speech API may recognize wrong words and the object detection can be off (since we use off-the-shelf VLM for zero-shot detection). We isolate the speech and hand detection errors when evaluating the system’s success rate and expect these can be resolved with better specialized systems.  We generally take two measures to account for other error modes. The first one is to make the system as transparent as possible so we can catch the error before it propagates to the next level (e.g. robot will announce the action it plans to take and confirm the target with the human user before it takes any action). The second measure is to catch exceptions in the code and detect simple failures (e.g. syntactic errors and grasp failures). For code error, the robot will announce its failure and ask the user to try again, for grasping failure the robot will adjust its motion primitive parameters and try again. While we do not directly tackle low-level action primitive failures, our system allows the user to fix low-level execution errors by incorporating directional pointing gestures. In one of our long-horizon tasks (Fig.4 in the paper), we show that the robot missed the grasping point of the mug handle by a bit and the user can point in the direction they want the robot to move in order to fix it.

---

> ### Author Response · Authors · 2023-08-12
> **Limitations of GIRAF (Reviewer zpDJ + Reviewer h9cL + Reviewer i78B)**
>
> We present further discussion of limitations of GIRAF below and add them in the paper (section 5):
>
> **Capabilities and Limitations of the framework**
> GIRAF is a system that enables additional gesture input alongside language instructions so it in principle can solve any language-only task with similar performance as baseline methods (e.g. Code as Policies). However, the ability to solve tasks with language-only instructions is often limited by the perception capability of the system. The lack of geometric reasoning capability in existing LLM and VLM greatly hinders these systems’ performance (e.g. to locate “an object on the 5th column and 3rd row”). GIRAF circumvents the geometric reasoning problem by incorporating human gestures for grounding physical information (e.g. “pick up this”,”move over here”). In this work we show a large performance improvement by just detecting pointing gestures in our user study. However, GIRAF as a framework still cannot solve tasks that require complex reasoning about the motion of the gestures, such as manipulative gestures (e.g. human user demonstrates a task and says “do this”), which is an interesting open problem for future work.
>
> **Limitations of our implementation in this work**
> In this work, the instantiation of GIRAF can only handle static gestures such as pointing, handover, thumbs-up/ thumbs-down, etc. While we show our reasoning module can handle dynamic gestures, we lack a model that can describe dynamic gestures accurately, so the full system cannot handle this type of gestures yet. One example that requires dynamic gestures is that the user performs the screwing motion to show the robot how to use a screwdriver. One can build a specialized dynamic gesture detector through collecting training data or leverage existing VLMs. While the SoTA VLMs we’ve tried (BLIP2, Instruct-BLIP, PaLI) do not take multiple images as input, we expect VLMs can understand/describe human dynamic gestures in the near future.
> To clarify, we have physical experiments more than just “pointing” but only did our user study with deictic pointing gestures to highlight the benefit of GIRAF for identifying referents in tasks where the goal can be hard to describe. We also show on the physical robot that GIRAF can perform long-horizon tasks with diverse types of gestures including pointing to object/location/direction, “palm up” (hand over), OK sign, and fist (see long-horizon tasks in the original paper -Fig.4 and in the Appendix - Fig.10).

---

### Decision · Program_Chairs · 2023-08-30

**Decision:**

Accept (Poster)

**Comment:**

Summary of the paper:

The paper presents GIRAF, which enhances human-robot interaction by integrating gestures with language instructions. GIRAF leverages Large Language Models (LLMs) with off-the-shelf perception models to interpret scene descriptors, recognize human gestures, and generate task plans accordingly. Evaluation on a real robot reveals the potential benefits of this method. Additionally, the authors introduces the GestureInstruct dataset, designed to examine the ability of LLMs to understand and categorize diverse gesture descriptions.

Strengths:
- Novel Integration: The paper's main strength is combining gestures with language instructions to design effective human-robot interaction.
- Practical Implications: The robot experiment underlines that gestures are an intuitive and preferred method of communication for users, potentially reshaping future HRI designs.
- Comprehensive System Design: The manuscript presents a holistic system, encompassing gesture identification, language processing, LLM-based planning, and execution of robotic tasks. A user study adds weight to its practical evaluation.
- Dataset Contribution: The GestureInstruct dataset is a promising tool for further research in gesture-based robotic instructions.
- Clarity & Resources: Reviewers appreciated the clear writing, informative discussions, and the helpful videos accompanying the paper.

Weaknesses:
- Role & Necessity of LLMs: Multiple reviewers expressed concerns regarding the clear definition and the actual need for LLMs in the context of gesture reasoning. Questions were raised about the LLM's contribution beyond gesture classification and task planning.
- Comparative Analysis: There's a perceived gap in comparing the proposed LLM-based method with other baselines, especially those which already account for pointing gestures. This makes it challenging to pinpoint the actual advantages of GIRAF.
- Evaluation Ambiguity: The quantitative results lack clear context regarding the metrics' meanings and the total tests performed, making it hard to interpret the evaluation's significance.
- Experimentation Limitations: Some experiments, like those involving only "pointing" gestures or where the human is very close to the object, seem contrived and don't necessarily reflect real-world scenarios.
- System Robustness: The approach's reliance on several existing models introduces potential error propagation. There are concerns about the robustness and safety of plans generated by LLMs, especially when individual modules fail.
- Contextual Nuances: The paper suggests gestures can help clarify language instructions, but there is a limited exploration on how language can clarify ambiguous gestures. Also, some presented situations, like "give me this tool," might seem artificially constructed without sufficient real-world applicability discussions.

Rebuttal and Discussion:

The authors responded to the questions from the reviewers thoughtfully and addressed several of the concerns raised. Furthermore, they submitted a revised manuscript reflecting the results of the discussion, and the quality of the paper improved with detailed data added to the appendix. However, not all concerns have been completely resolved, and the revised manuscript exceeds the page limit. Therefore, further revisions are required.

Recommendation:

Given the novel integration of gestures and language for HRI, GIRAF's proposition is promising. However, for a clearer evaluation of its significance and a broader acceptance in the community, it is imperative to address the stated concerns, particularly around the precise role of LLMs, comparative analyses with baselines, and the system's robustness in diverse scenarios.